# FedBE: Making Bayesian Model Ensemble Applicable to Federated Learning

**Hong-You Chen**
The Ohio State University, USA
chen.9301@osu.edu

**Wei-Lun Chao**
The Ohio State University, USA
chao.209@osu.edu

## Abstract

Federated learning aims to collaboratively train a strong global model by accessing users' locally trained models but not their own data. A crucial step is therefore to aggregate local models into a global model, which has been shown challenging when users have non-i.i.d. data. In this paper, we propose a novel aggregation algorithm named FEDBE, which takes a Bayesian inference perspective by sampling higher-quality global models and combining them via **B**ayesian model **E**nsemble, leading to much robust aggregation. We show that an effective model distribution can be constructed by simply fitting a Gaussian or Dirichlet distribution to the local models. Our empirical studies validate FEDBE's superior performance, especially when users' data are not i.i.d. and when the neural networks go deeper. Moreover, FEDBE is compatible with recent efforts in regularizing users' model training, making it an easily applicable module: you only need to replace the aggregation method but leave other parts of your federated learning algorithm intact.

## 1 Introduction

Modern machine learning algorithms are data and computation hungry. It is therefore desired to collect as many data and computational resources as possible, for example, from individual users (e.g., users' smartphones and pictures taken on them), without raising concerns in data security and privacy. Federated learning has thus emerged as a promising learning paradigm, which leverages individuals' computational powers and data securely — by only sharing their locally trained models with the server — to jointly optimize a global model (Konečný et al., 2016; Yang et al., 2019).

Federated learning (FL) generally involves multiple rounds of communication between the server and clients (i.e., individual sites). Within each round, the clients first train their own models using their own data, usually with limited sizes. The server then aggregates these models into a single, global model. The clients then begin the next round of training, using the global model as the initialization.

We focus on **model aggregation**, one of the most critical steps in FL. The standard method is FEDAVG (McMahan et al., 2017), which performs element-wise average over clients' model weights. Assuming that each client's data are sampled i.i.d. from their aggregated data, FEDAVG has been shown convergent to the *ideal* model trained in a centralized way using the aggregated data (Zinkevich et al., 2010; McMahan et al., 2017; Zhou & Cong, 2017). Its performance, however, can degrade drastically if such an assumption does not hold in practice (Karimireddy et al., 2020; Li et al., 2020b; Zhao et al., 2018): FEDAVG simply drifts away from the ideal model. Moreover, by only taking weight average, FEDAVG does not fully utilize the information among clients (e.g., variances), and may have negative effects on over-parameterized models like neural networks due to their permutation-invariant property in the weight space (Wang et al., 2020; Yurochkin et al., 2019).

To address these issues, we propose a novel aggregation approach using *Bayesian inference*, inspired by (Maddox et al., 2019). Treating each client's model as a possible global model, we construct a distribution of global models, from which weight average (i.e., FEDAVG) is one particular sample and many other global models can be sampled. This distribution enables Bayesian model ensemble — aggregating the outputs of a wide spectrum of global models for a more robust prediction. We show that Bayesian model ensemble can make more accurate predictions than weight average at a single round of communication, especially under the non i.i.d. client condition. Nevertheless, lacking

a *single* global model that represents Bayesian model ensemble and can be sent back to clients, Bayesian model ensemble cannot directly benefit federated learning in a multi-round setting.

We therefore present FEDBE, a learning algorithm that effectively incorporates **B**ayesian model **E**nsemble into federated learning. Following (Guha et al., 2019), we assume that the server has access to a set of unlabeled data, on which we can make predictions by model ensemble. This assumption can easily be satisfied: the server usually collects its own data for model validation, and collecting unlabeled data is simpler than labeled ones. (See section 6 for more discussion, including the privacy concern.) Treating the ensemble predictions as the "pseudo-labels" of the unlabeled data, we can then summarize model ensemble into a single global model by knowledge distillation (Hinton et al., 2015) — using the predicted labels (or probabilities or logits) as the *teacher* to train a *student* global model. The student global model can then be sent back to the clients to begin their next round of training[1].

We identify one key detail of knowledge distillation in FEDBE. In contrast to its common practice where the teacher is highly accurate and labeled data are accessible, the ensemble predictions in federated learning can be relatively *noisy*[2]. To prevent the student from over-fitting the noise, we apply stochastic weight average (SWA) (Izmailov et al., 2018) in distillation. SWA runs stochastic gradient descent (SGD) with a *cyclical* learning rate and averages the weights of the traversed models, allowing the traversed models to jump out of noisy local minimums, leading to a more robust student.

We validate FEDBE on CIFAR-10/100 (Krizhevsky et al., 2009) and Tiny-ImageNet (Le & Yang, 2015) under different client conditions (i.e., i.i.d. and non-i.i.d. ones), using ConvNet (TensorFlow team, 2016), ResNet (He et al., 2016), and MobileNetV2 (Howard et al., 2017; Sandler et al., 2018). FEDBE consistently outperforms FEDAVG, especially when the neural network architecture goes deeper. Moreover, FEDBE can be *compatible* with existing FL algorithms that regularize clients' learning or leverage server momentum (Li et al., 2020a; Sahu et al., 2018; Karimireddy et al., 2020; Hsu et al., 2019) and further improves upon them. Interestingly, even if the unlabeled server data have a different distribution or domain from the test data (e.g., taken from a different dataset), FEDBE can still maintain its accuracy, making it highly applicable in practice.

## 2 RELATED WORK (MORE IN APPENDIX A)

**Federated learning (FL).** In the multi-round setting, FEDAVG (McMahan et al., 2017) is the standard approach. Many works have studied its effectiveness and limitation regarding convergence, robustness, and communication cost, especially in the situations of non-i.i.d. clients. Please see Appendix A for a list of works. Many works proposed to improve FEDAVG. FEDPROX (Li et al., 2020a; Sahu et al., 2018), FEDDANE (Li et al., 2019), Yao et al. (2019), and SCAFFOLD (Karimireddy et al., 2020) designed better local training strategies to prevent clients' model drifts. Zhao et al. (2018) studied the use of shared data between the server and clients to reduce model drifts. Reddi et al. (2020) and Hsu et al. (2019) designed better update rules for the global model by server momentum and adaptive optimization. Our FEDBE is complementary to and can be compatible with these efforts.

In terms of model aggregation. Yurochkin et al. (2019) developed a Bayesian non-parametric approach to match clients' weights before average, and FEDMA (Wang et al., 2020) improved upon it by iterative layer-wise matching. One drawback of FEDMA is its linear dependence of computation and communication on the network's depth, not suitable for deeper models. Also, both methods are not yet applicable to networks with residual links and batch normalization (Ioffe & Szegedy, 2015). We improve aggregation via Bayesian ensemble and knowledge distillation, bypassing weight matching.

**Ensemble learning and knowledge distillation.** Model ensemble is known to be more robust and accurate than individual base models (Zhou, 2012; Dietterich, 2000; Breiman, 1996). Several recent works (Anil et al., 2018; Guo et al., 2020; Chen et al., 2020) investigated the use of model ensemble and knowledge distillation (Hinton et al., 2015) in an online fashion to jointly learn multiple models, where the base models and distillation have access to the centralized labeled data or decentralized data of the same distribution. In contrast, client models in FL are learned with isolated and likely

---

[1]Distillation from the ensemble of clients' models was explored in (Guha et al., 2019) for a one-round federated setting. Our work can be viewed as an extension to the multi-round setting, by sampling more and higher-quality models as the bases for more robust ensemble.

[2]We note that, the ensemble predictions can be noisy but still more accurate than weight average (see Figure 3 and subsection C.2).

non-i.i.d. and limited data; our distillation is performed without labeled data. We thus propose to sample base models of higher quality for Bayesian ensemble and employ SWA for robust distillation.

**Knowledge distillation in FL.** Guha et al. (2019) considered *one-round* FL and applied distillation to obtain a global model from the *direct ensemble of clients' models*. A similar idea was used in (Papernot et al., 2017) in a different context. Our method can be viewed as an extension of (Guha et al., 2019) to multi-round FL, with higher-quality base models being sampled from a global distribution for more robust ensemble. Knowledge distillation was also used in (Li & Wang, 2019) and (Jeong et al., 2018) but for different purposes. Li & Wang (2019) performed ensemble distillation for each client, aiming to learn strong personalized models but not the global model. Jeong et al. (2018) aimed to speed up communication by sending averaged logits of clients' data, not models, between clients and the server. The clients then use the aggregated logits to regularize local training via distillation. The accuracy, however, drops drastically compared to FEDAVG in exchange for faster communication. In contrast, we distill on the server using unlabeled data collected at the server, aiming to build a stronger global model. The most similar work to ours is a concurrent work by Lin et al. (2020)[3], which also employs ensemble distillation on the server in a multi-round setting. *Our work is notably different from all the above methods by taking the Bayesian perspective to sample better base models and investigating* SWA *for distillation, significantly improving the performance on multi-round FL.*

## 3 BAYESIAN MODEL ENSEMBLE FOR FEDERATED LEARNING

### 3.1 BACKGROUND: FEDAVG

Federated learning (FL) usually involves a server coordinating with many clients to jointly learn a global model without data sharing, in which FEDAVG (McMahan et al., 2017) in a standard approach. Denote by $\mathcal{S}$ the set of clients, $\mathcal{D}_i = \{(\boldsymbol{x}_n, y_n)\}_{n=1}^{N_i}$ the labeled data of client $i$, and $\bar{\boldsymbol{w}}$ the weights of the current global model, FEDAVG starts with **client training** of all the clients in parallel, initializing each clients' model $\boldsymbol{w}_i$ with $\bar{\boldsymbol{w}}$ and performing SGD for $K$ steps with a step size $\eta_l$

$$\text{Client training:} \qquad \boldsymbol{w}_i \leftarrow \boldsymbol{w}_i - \eta_l \nabla \ell(B_k, \boldsymbol{w}_i), \text{ for } k = 1, 2, \cdots, K, \qquad (1)$$

where $\ell$ is a loss function and $B_k$ is the mini-batch sampled from $\mathcal{D}_i$ at the $k$th step. After receiving all the clients' models $\{\boldsymbol{w}_i; i \in \mathcal{S}\}$, given $|\mathcal{D}| = \sum_i |\mathcal{D}_i|$, FEDAVG performs weight average to update the global model $\bar{\boldsymbol{w}}$

$$\text{Model aggregation (by weight average):} \qquad \bar{\boldsymbol{w}} \leftarrow \sum_i \frac{|\mathcal{D}_i|}{|\mathcal{D}|} \boldsymbol{w}_i. \qquad (2)$$

With the updated global model $\bar{\boldsymbol{w}}$, FEDAVG then starts the next round of **client training**. The whole procedure of FEDAVG therefore iterates between Equation 1 and Equation 2, for $R$ rounds.

In the case that $\mathcal{D}_i$ is i.i.d. sampled from the aggregated data $\mathcal{D} = \bigcup_{i \in \mathcal{S}} \mathcal{D}_i$, FEDAVG has been shown convergent to the *ideal* model $\boldsymbol{w}^\star$ learned directly from $\mathcal{D}$ in a centralized manner (Stich, 2019; Haddadpour & Mahdavi, 2019; Khaled et al., 2020). In reality, however, the server has little control and knowledge about the clients. Each client may have different data distributions in the input (e.g., image distribution) or output (e.g., label distribution). Some clients may disconnect at certain rounds. All of these factors suggest the non-i.i.d. nature of federated learning in practice, under which the effectiveness of FEDAVG can largely degrade (Zhao et al., 2018; Li et al., 2020b; Hsu et al., 2019). For example, Karimireddy et al. (2020) show that $\bar{\boldsymbol{w}}$ in Equation 2 can drift away from $\boldsymbol{w}^\star$.

### 3.2 A BAYESIAN PERSPECTIVE

We propose to view the problem of model drift from a Bayesian perspective. In Bayesian learning, it is the posterior distribution $p(\boldsymbol{w}|\mathcal{D})$ of the global model being learned, from which $\bar{\boldsymbol{w}}$ and $\boldsymbol{w}^\star$ can be regarded as two particular samples (i.e., point estimates). Denote by $p(y|\boldsymbol{x}; \boldsymbol{w})$ the output probability of a global model $\boldsymbol{w}$, one approach to mitigate model drift is to perform Bayesian inference (Neal, 2012; Barber, 2012) for prediction, integrating the outputs of all possible models w.r.t. the posterior

$$p(y|\boldsymbol{x}; \mathcal{D}) = \int p(y|\boldsymbol{x}; \boldsymbol{w}) p(\boldsymbol{w}|\mathcal{D}) d\boldsymbol{w} \qquad (3)$$

---

[3]We notice a generalized version of it in (He et al., 2020) that improves the computational efficiency.

rather than relying on a single point estimate. While Equation 3 is intractable in general, we can approximate it by the Monte Carlo method, sampling $M$ models for model ensemble

$$\textbf{Bayesian model ensemble:} \quad p(y|\boldsymbol{x}; \mathcal{D}) \approx \frac{1}{M} \sum_{m=1}^{M} p(y|\boldsymbol{x}; \boldsymbol{w}^{(m)}), \text{ where } \boldsymbol{w}^{(m)} \sim p(\boldsymbol{w}|\mathcal{D}). \quad (4)$$

The question is: how to estimate $p(\boldsymbol{w}|\mathcal{D})$ in federated learning, given merely client models $\{\boldsymbol{w}_i\}$?

### 3.3 BAYESIAN MODEL ENSEMBLE WITH APPROXIMATED POSTERIORS

We resort to a recently proposed idea, named stochastic weight average-Gaussian (SWAG) (Maddox et al., 2019), for estimating the posterior. SWAG employed a cyclical or constant learning rate in SGD, following SWA (Izmailov et al., 2018). SWAG then constructs a Gaussian distribution $p(\boldsymbol{w}|\mathcal{D})$ by fitting the parameters to the model weights it traverses in SGD.

In federated learning, by rewriting $\boldsymbol{w}_i$ as $\bar{\boldsymbol{w}} - \boldsymbol{g}_i$, where $\boldsymbol{g}_i = -(\boldsymbol{w}_i - \bar{\boldsymbol{w}})$ denotes the $K$-step stochastic gradient on a mini-batch $\mathcal{D}_i \subset \mathcal{D}$ (McMahan et al., 2017), we can indeed view each client's model $\boldsymbol{w}_i$ as taking $K$-step SGD to traverse the weight space of global models.

**Gaussian.** To this end, we propose to fit a diagonal Gaussian distribution $\mathcal{N}(\boldsymbol{\mu}, \Sigma_{\text{diag}})$ to the clients' models $\{\boldsymbol{w}_i\}$ following (Maddox et al., 2019),

$$\boldsymbol{\mu} = \sum_i \frac{|\mathcal{D}_i|}{|\mathcal{D}|} \boldsymbol{w}_i, \qquad \Sigma_{\text{diag}} = \text{diag} \left( \sum_i \frac{|\mathcal{D}_i|}{|\mathcal{D}|} (\boldsymbol{w}_i - \boldsymbol{\mu})^2 \right), \quad (5)$$

from which we can sample $\{\boldsymbol{w}^{(m)} \sim \mathcal{N}(\boldsymbol{\mu}, \Sigma_{\text{diag}})\}_{m=1}^{M}$ for model ensemble (cf. Equation 4). Here $(\cdot)^2$ means taking element-wise square. We note that, both the clients' models $\{\boldsymbol{w}_i\}$ and FEDAVG $\bar{\boldsymbol{w}}$ are possible samples from $\mathcal{N}(\boldsymbol{\mu}, \Sigma_{\text{diag}})$.

**Dirichlet.** We investigate another way to construct $p(\boldsymbol{w}|\mathcal{D})$, inspired by the fact that an averaged stochastic gradient is in general closer to the true gradient than individual stochastic gradients (Haddadpour & Mahdavi, 2019; Izmailov et al., 2018; Liang et al., 2019; Stich, 2019; Zhou & Cong, 2017). By viewing each client's model as $\boldsymbol{w}_i = \bar{\boldsymbol{w}} - \boldsymbol{g}_i$, such a fact suggests that a convex combination (i.e., weighted average) of clients' models can lead to a better model than each client alone:

$$\boldsymbol{w} = \sum_i \frac{\gamma_i |\mathcal{D}_i|}{\sum_{i'} \gamma_{i'} |\mathcal{D}_{i'}|} \boldsymbol{w}_i = \bar{\boldsymbol{w}} - \sum_i \frac{\gamma_i |\mathcal{D}_i|}{\sum_{i'} \gamma_{i'} |\mathcal{D}_{i'}|} \boldsymbol{g}_i, \quad (6)$$

where $\boldsymbol{\gamma} = [\gamma_1, \cdots, \gamma_{|\mathcal{S}|}]^\top \in \Delta^{|\mathcal{S}|-1}$ is a vector on the $(|\mathcal{S}| - 1)$-simplex. To this end, we use a Dirichlet distribution $\text{Dir}(\boldsymbol{\alpha})$ to model the distribution of $\boldsymbol{\gamma}$, from which we can then sample $\boldsymbol{w}^{(m)}$ by

$$\boldsymbol{w}^{(m)} = \sum_i \frac{\gamma_i^{(m)} |\mathcal{D}_i|}{\sum_{i'} \gamma_{i'}^{(m)} |\mathcal{D}_{i'}|} \boldsymbol{w}_i, \qquad \boldsymbol{\gamma}^{(m)} \sim p(\boldsymbol{\gamma}) = p(\gamma_1, \cdots, \gamma_{|\mathcal{S}|}) = \frac{1}{\text{B}(\boldsymbol{\alpha})} \prod_i \gamma_i^{\alpha_i - 1}, \quad (7)$$

where $\boldsymbol{\alpha} = [\alpha_1, \cdots, \alpha_{|\mathcal{S}|}]^\top \succ \boldsymbol{0}$ is the parameter of a Dirichlet distribution, and $\text{B}(\boldsymbol{\alpha})$ is the multivariate beta function for normalization. We study different $\boldsymbol{\alpha}$ in subsection C.1.

To sum up, Bayesian model ensemble in federated learning takes the following two steps:

- Construct $p(\boldsymbol{w}|\mathcal{D})$ from the clients' models $\{\boldsymbol{w}_i\}$ (cf. Equation 5 or Equation 7)
- Sample $\{\boldsymbol{w}^{(m)} \sim p(\boldsymbol{w}|\mathcal{D})\}_{m=1}^{M}$ and perform ensemble (cf. Equation 4)

**Analysis.** We validate Bayesian model ensemble with a three-class classification problem on the Swiss roll data in Figure 1 (a). We consider three clients with the same amount of training data: each has $80\%$ data from one class and $20\%$ from the other two classes, essentially a non-i.i.d. case. We apply FEDAVG to train a two-layer MLP for 10 rounds (each round with 2 epochs). We then show the test accuracy of models sampled from Equation 7 (with $\boldsymbol{\alpha} = 0.5 \times \boldsymbol{1}$) — the corners of the triangle (i.e., $\Delta^2$) in Figure 1 (b) correspond to the clients; the position inside the triangle corresponds to the $\boldsymbol{\gamma}$ coefficients. We see that, the sampled models within the triangle usually have higher accuracy than the clients' models. Surprisingly, the best performing model that can be sampled from a Dirichlet distribution is not FEDAVG (the center of the triangle), but the one drifting to the bottom right. This suggests that Bayesian model ensemble can lead to higher accuracy (by averaging over sampled

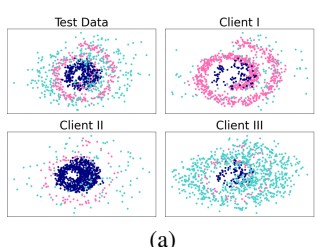 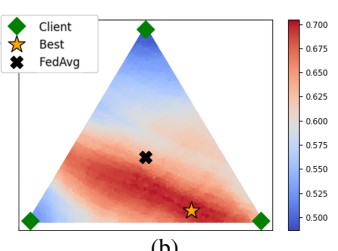 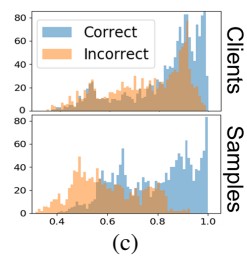

| (a) | (b) | (c) |

Figure 1: **An illustration of models that can be sampled from a Dirichlet distribution (Equation 7).** (a) A three-class toy data with three clients, each has non-i.i.d. imbalanced data. (b) We show the sampled model's corresponding $\boldsymbol{\gamma}$ **(position in the triangle)** and its **test accuracy (color)**. FEDAVG is at the center; clients' models are at corners. The best performing model (star) is not at the center, drifting away from FEDAVG. (c) Histograms of (in)correctly predicted examples at different confidences (x-axis) by sampled models and clients.

models) than FEDAVG alone. Indeed, by combining 10 randomly sampled models via Equation 4, Bayesian model ensemble attains a 69% test accuracy, higher than 64% by FEDAVG. Figure 1 (c) further shows that the sampled models have a better alignment between the prediction confidence and accuracy than the clients' (mean results of 3 clients or samples). See subsection C.1 for details.

To further compare FEDAVG and ensemble, we linearize $p(y|\boldsymbol{x}; \cdot)$ at $\bar{\boldsymbol{w}}$ (Izmailov et al., 2018),

$$p(y|\boldsymbol{x}; \boldsymbol{w}^{(m)}) = p(y|\boldsymbol{x}; \bar{\boldsymbol{w}}) + \langle \nabla p(y|\boldsymbol{x}; \bar{\boldsymbol{w}}), \Omega^{(m)} \rangle + O(\|\Omega^{(m)}\|^2), \tag{8}$$

where $\Omega^{(m)} = \boldsymbol{w}^{(m)} - \bar{\boldsymbol{w}}$ and $\langle \cdot, \cdot \rangle$ is the dot product. By averaging the sampled models, we arrive at

$$\frac{1}{M} \sum_m p(y|\boldsymbol{x}; \boldsymbol{w}^{(m)}) - p(y|\boldsymbol{x}; \bar{\boldsymbol{w}}) = \langle \nabla p(y|\boldsymbol{x}; \bar{\boldsymbol{w}}), \frac{1}{M} \sum_m \Omega^{(m)} \rangle + O(\Omega^2) = O(\Omega^2), \tag{9}$$

where $\Omega = \max_m \|\Omega^{(m)}\|$. In federated learning, especially in the non-i.i.d. cases, $\Omega$ can be quite large. Bayesian ensemble thus can have a notable difference (improvement) compared to FEDAVG.

## 4 FEDBE

Bayesian model ensemble, however, cannot directly benefit multi-round federated learning, in which a *single* global model must be sent back to the clients to continue **client training**. We must *translate* the prediction rule of Bayesian model ensemble into a single global model.

To this end, we make an assumption that we can access a set of *unlabeled* data $\mathcal{U} = \{\boldsymbol{x}_j\}_{j=1}^J$ at the server. This can easily be satisfied since collecting unlabeled data is simpler than labeled ones. We use $\mathcal{U}$ for two purposes. On one hand, we use $\mathcal{U}$ to *memorize* the prediction rule of Bayesian model ensemble, turning $\mathcal{U}$ into a pseudo-labeled set $\mathcal{T} = \{(\boldsymbol{x}_j, \hat{\boldsymbol{p}}_j)\}_{j=1}^J$, where $\hat{\boldsymbol{p}}_j = \frac{1}{M} \sum_{m=1}^M p(y|\boldsymbol{x}_j; \boldsymbol{w}^{(m)})$ is a probability vector. On the other hand, we use $\mathcal{T}$ as supervision to train a global model $\boldsymbol{w}$, aiming to *mimic* the prediction rule of Bayesian model ensemble on $\mathcal{U}$.

---

**Algorithm 1:** FEDBE  (**Fed**erated **B**ayesian **E**nsemble)

**Server input** : initial global model $\boldsymbol{w}$, SWA scheduler $\eta_{\text{SWA}}$, unlabeled data $\mathcal{U} = \{\boldsymbol{x}_j\}_{j=1}^J$;

**Client $i$'s input:** local step size $\eta_l$, local labeled data $\mathcal{D}_i$;

**for** $r \leftarrow 1$ **to** $R$ **do**
    **Sample** clients $\mathcal{S} \subseteq \{1, \cdots, N\}$;
    **Communicate** $\boldsymbol{w}$ to all clients $i \in \mathcal{S}$;
    **for** *each client $i \in \mathcal{S}$ in parallel* **do**
        **Initialize** local model $\boldsymbol{w}_i \leftarrow \boldsymbol{w}$;
        $\boldsymbol{w}_i \leftarrow$ **Client training**$(\boldsymbol{w}_i, \mathcal{D}_i, \eta_l)$; [Equation 1]
        **Communicate** $\boldsymbol{w}_i$ to the server;
    **end**
    **Construct** $\bar{\boldsymbol{w}} = \sum_{i \in \mathcal{S}} \frac{|\mathcal{D}_i|}{\sum_{i' \in \mathcal{S}} |\mathcal{D}'_i|} \boldsymbol{w}_i$;
    **Construct** global model distribution $p(\boldsymbol{w}|\mathcal{D})$ from $\{\boldsymbol{w}_i; i \in \mathcal{S}\}$; [Equation 5 or Equation 7]
    **Sample** $M$ global models $\{\boldsymbol{w}^{(m)} \sim p(\boldsymbol{w}|\mathcal{D})\}_{m=1}^M$;
    **Construct** $\{\boldsymbol{w}^{(m')}\}_{m'=1}^{M'} = \{\bar{\boldsymbol{w}}\} \cup \{\boldsymbol{w}_i; i \in \mathcal{S}\} \cup \{\boldsymbol{w}^{(m)}\}_{m=1}^M$;
    **Construct** $\mathcal{T} = \{\boldsymbol{x}_j, \hat{\boldsymbol{p}}_j\}_{j=1}^J$, where $\hat{\boldsymbol{p}}_j = \frac{1}{M'} \sum_{m'} p(y|\boldsymbol{x}_j; \boldsymbol{w}^{(m')})$; [Equation 4]
    **Knowledge distillation:** $\boldsymbol{w} \leftarrow \text{SWA}(\bar{\boldsymbol{w}}, \mathcal{T}, \eta_{\text{SWA}})$;
**end**

**Server output** : $\boldsymbol{w}$.

---

This process is reminiscent of *knowledge distillation* (Hinton et al., 2015) to transfer knowledge from a teacher model (in our case, the Bayesian model ensemble) to a student model (a single global model). Here we apply a cross entropy loss to learn $\boldsymbol{w}$ : $-\frac{1}{J} \sum_j \hat{\boldsymbol{p}}_j^\top \log(p(y|\boldsymbol{x}_j; \boldsymbol{w}))$.

**SWA for knowledge distillation.** Optimizing $\boldsymbol{w}$ using standard SGD, however, may arrive at a suboptimal solution: the resulting $\boldsymbol{w}$ can have much worse test accuracy than ensemble. We identify

one major reason: the ensemble prediction $\hat{\boldsymbol{p}}_j$ can be noisy (e.g., $\arg\max_c \hat{p}_j[c]$ is not the true label of $\boldsymbol{x}_j$), especially in the early rounds of FL. The student model $\boldsymbol{w}$ thus may over-fit the noise. We note that, this finding does not contradict our observations in subsection 3.3: Bayesian model ensemble has higher test accuracy than FEDAVG but is still far from being perfect (i.e., $100\%$ accuracy). To address this issue, we apply SWA (Izmailov et al., 2018) to train $\boldsymbol{w}$. SWA employs a cyclical learning rate schedule in SGD by periodically imposing a sharp increase in step sizes and averages the weights of models it traverses, enabling $\boldsymbol{w}$ to jump out of noisy local minimums. As will be shown in section 5, SWA consistently outperforms SGD in distilling the ensemble predictions into the global model.

We name our algorithm FEDBE (**Fed**erated **B**ayesian **E**nsemble) and summarize it in algorithm 1. While knowledge distillation needs extra computation at the server, it is hardly a concern as the server is likely computationally rich. (See subsection D.1 for details.) We also empirically show that a small number of sampled models (e.g., $M = 10 \sim 20$) are already sufficient for FEDBE to be effective.

## 5 EXPERIMENT

### 5.1 SETUP (MORE DETAILS IN APPENDIX B)

**Datasets, models, and settings.** We use CIFAR-10/100 (Krizhevsky et al., 2009), both contain 50K training and 10K test images, from 10 and 100 classes. We also use Tiny-ImageNet (Le & Yang, 2015), which has 500 training and 50 test images per class for 200 classes. We follow (McMahan et al., 2017) to use a ConvNet (LeCun et al., 1998) with 3 convolutional and 2 fully-connected layers. We also use ResNet-{20, 32, 44, 56} (He et al., 2016) and MobileNetV2 (Sandler et al., 2018). We split part of the training data to the server as the unlabeled data, distribute the rest to the clients, and evaluate on the test set. We report mean $\pm$ standard deviation (std) over five times of experiments.

**Implementation details.** As mentioned in (McMahan et al., 2017; Wang et al., 2020; Li et al., 2020b), FEDAVG is sensitive to the local training epochs $E$ per round ($E = \lceil \frac{K|B_K|}{|\mathcal{D}_i|} \rceil$ in Equation 1). Thus, in each experiment, we first tune $E$ from $[1, 5, 10, 20, 30, 40]$ for FEDAVG and adopt the same $E$ to FEDBE. Li et al. (2020b); Reddi et al. (2020) suggested that the local step size $\eta_l$ (see Equation 1) must decay along the communication rounds in non-i.i.d. settings for convergence. We set the initial $\eta_l$ as 0.01 and decay it by 0.1 at 30% and 60% of total rounds, respectively. Within each round of local training, we use SGD optimizer with weight decay and a 0.9 momentum and impose no further decay on local step sizes. Weight decay is crucial in local training (cf. subsection B.3). For ResNet and MobileNetV2, we use batch normalization (BN). See subsection C.5 for a discussion on using group normalization (GN) (Wu & He, 2018; Hsieh et al., 2020), which converges much slower.

**Baselines.** Besides FEDAVG, we compare to one-round training with 200 local epochs followed by model ensemble at the end (**1-Ensemble**). We also compare to vanilla knowledge distillation (**v-Distillation**), which performs ensemble directly over clients' models and uses a SGD momentum optimizer (with a batch size of 128 for 20 epochs) for distillation in each round. For fast convergence, we initialize distillation with the weight average of clients' models and sharpen the pseudo label as $\hat{p}_j[c] \leftarrow \hat{p}_j[c]^2 / \sum_{c'} \hat{p}_j[c']^2$, similar to (Berthelot et al., 2019). We note that, **v-Distillation** is highly similar to (Lin et al., 2020) except for different hyper-parameters. We also compare to FEDPROX (Li et al., 2020a) and FEDAVGM (Hsu et al., 2019) on better local training and using server momentum.

**FEDBE.** We focus on **Gaussian** (cf. Equation 5). Results with **Dirichlet** distributions are in subsection C.1. We sample ***M=10*** models and combine them with the weight average of clients and individual clients for ensemble. For distillation, we apply SWA (Izmailov et al., 2018), which uses a cyclical schedule with the step size $\eta_{\text{SWA}}$ decaying from $1\text{e}{-}3$ to $4\text{e}{-}4$, and collect models at the end of every cycle (every 25 steps) after the 250th step. We follow other settings of v-Distillation (e.g., distill for 20 epochs per round). We average the collected models to obtain the global model.

### 5.2 MAIN STUDIES: CIFAR-10 WITH NON-I.I.D. CLIENTS USING DEEP NEURAL NETWORKS

**Setup.** We focus on CIFAR-10. We randomly split 10K training images to be the unlabeled data at the server. We distribute the remaining images to **10** clients with two non-i.i.d. cases. **Step:** Each client has 8 minor classes with 10 images per class, and 2 major classes with 1,960 images per class, inspired by (Cao et al., 2019). **Dirichlet:** We follow (Hsu et al., 2019) to simulate a heterogeneous

Table 1: Mean±std of test accuracy (%) on non-i.i.d. CIFAR-10. ★: trained with 50K images without splitting.

| Non-i.i.d. Type | Method | ConvNet | ResNet20 | ResNet32 | ResNet44 | ResNet56 |
|---|---|---|---|---|---|---|
| Step | 1-Ensemble | 60.5±0.28 | 49.9±0.46 | 35.5±0.50 | 32.8±0.38 | 23.3±0.52 |
| | FEDAVG | 72.0±0.25 | 70.2±0.17 | 66.5±0.36 | 60.5±0.26 | 51.4±0.15 |
| | v-Distillation | 69.2±0.18 | 72.6±0.62 | 68.4±0.33 | 60.4±0.53 | 56.4±1.10 |
| | FEDBE (w/o SWA) | 72.1±1.21 | 74.9±1.41 | 71.1±0.75 | 61.0±0.75 | 56.6±0.85 |
| | **FEDBE** | **74.5±0.51** | **77.5±0.42** | **72.7±0.27** | **65.5±0.32** | **60.7±0.45** |
| Dirichlet | 1-Ensemble | 63.3±0.56 | 45.2±1.06 | 39.5±0.78 | 31.5±0.77 | 27.2±0.65 |
| | FEDAVG | 72.3±0.12 | 74.4±0.36 | 73.4±0.23 | 67.1±0.54 | 62.2±0.45 |
| | v-Distillation | 67.7±0.98 | 73.1±0.78 | 70.8±0.64 | 66.9±0.85 | 62.8±0.66 |
| | FEDBE (w/o SWA) | 70.1±0.42 | 75.9±0.56 | 73.9±0.55 | 68.2±0.72 | 63.2±0.71 |
| | **FEDBE** | **73.9±0.45** | **78.2±0.36** | **77.7±0.45** | **71.5±0.38** | **67.0±0.30** |
| Centralized★ | SGD | 84.5 | 91.7 | 92.6 | 93.1 | 93.4 |

Table 2: Compatibility of FEDBE with FEDAVGM and FEDPROX on non-i.i.d. CIFAR-10.

| Non-i.i.d. Type | Method | ConvNet | ResNet20 | ResNet32 | ResNet44 | ResNet56 |
|---|---|---|---|---|---|---|
| Step | FEDPROX | 72.5±0.71 | 71.1±0.52 | 67.7±0.26 | 60.4±0.71 | 54.9±0.66 |
| | **FEDBE** +FEDPROX | **74.9±0.38** | **77.7±0.45** | **72.9±0.44** | **64.5±0.37** | **60.1±0.62** |
| | FEDAVGM | 72.3±0.55 | 73.2±0.57 | 70.0±0.62 | 59.9±0.65 | 52.7±0.49 |
| | **FEDBE** +FEDAVGM | **74.5±0.47** | **78.0±0.46** | **73.6±0.50** | **65.5±0.40** | **59.7±0.51** |
| Dirichlet | FEDPROX | 72.6±0.38 | 76.1±0.49 | 73.4±0.51 | 68.1±0.79 | 60.9±0.46 |
| | **FEDBE** +FEDPROX | **74.6±0.35** | **78.7±0.49** | **77.3±0.60** | **71.7±0.43** | **66.5±0.41** |
| | FEDAVGM | 73.0±0.43 | 76.5±0.44 | 75.5±0.79 | 67.7±0.46 | 58.9±0.72 |
| | **FEDBE** +FEDAVGM | **74.4±0.49** | **78.5±0.66** | **78.5±0.26** | **72.0±0.51** | **67.0±0.55** |

partition for $N$ clients on $C$ classes. For class $c$, we draw a $N$-dim vector $\boldsymbol{q}_c$ from $\mathrm{Dir}(0.1)$ and assign data to client $n$ proportionally to $\boldsymbol{q}_c[n]$. The clients have different numbers of total images.

**Results.** We implement all methods with $40$ rounds[4], except for the one-round Ensemble. We assume that all clients are connected at every round. We set the local batch size as $40$. Table 1 summarizes the results. FEDBE outperforms the baselines by a notable margin. Compared to FEDAVG, FEDBE consistently leads to a $2 \sim 9\%$ gain, which becomes larger as the network goes deeper. By comparing FEDBE to FEDBE (w/o SWA) and v-Distillation, we see the consistent improvement by SWA for distillation and Bayesian ensemble with sampled models. We note that, FEDAVG outperforms 1-Ensemble and is on a par with v-Distillation[5], justifying (a) the importance of multi-round training; (b) the challenge of ensemble distillation. *Please see subsection C.2 for an insightful analysis.*

**Compatibility with existing efforts.** Our improvement in model aggregation is compatible with recent efforts in better local training (Li et al., 2020a; Karimireddy et al., 2020) and using server momentum (Reddi et al., 2020; Hsu et al., 2019). Specifically, Reddi et al. (2020); Hsu et al. (2019) applied the server momentum to FEDAVG by treating FEDAVG in each round as a step of adaptive optimization. FEDBE can incorporate this idea by initializing distillation with their FEDAVG. Table 2 shows the results of FEDPROX (Li et al., 2020a) and FEDAVGM (Hsu et al., 2019), w/ or w/o FEDBE. FEDBE can largely improve them. The combination even outperforms FEDBE alone in many cases.

**Effects of Bayesian Ensemble.** We focus on the **Step** setting. We compare different combinations of client models **C**: $\{\boldsymbol{w}_i\}$, client weight average **A**: $\bar{\boldsymbol{w}}$, and $M$ samples from Gaussian **S**: $\{\boldsymbol{w}^{(m)}\}_{m=1}^{M}$ to construct the distillation targets $\mathcal{T}$ for FEDBE in algorithm 1. As shown in Table 3, sampling global models for Bayesian ensemble improves the accuracy. Sampling $M = 10 \sim 20$ samples (plus weight average and clients to form ensemble) is sufficient to make FEDBE effective (see Figure 2).

Table 3: FEDBE distillation targets. **A**: client average; **C**: clients; **S**: samples.

| Distillation Targets | ConvNet | ResNet20 |
|---|---|---|
| **A** | 72.6±0.28 | 73.4±0.46 |
| **S** | 73.1±0.46 | 75.2±0.61 |
| **S** + **A** | 73.9±0.33 | 76.1±0.47 |
| **A** + **C** | 73.0±0.36 | 75.4±0.38 |
| **S** + **C** | 74.0±0.66 | **77.9±0.56** |
| **S** + **A** + **C** | **74.5±0.51** | 77.5±0.42 |
| Ground-truth labels | 76.6±0.21 | 80.2±0.23 |

[4] We observe very little gain after 40 rounds: adding 60 rounds only improves FEDAVG (ConvNet) by $0.7\%$.
[5] In contrast to Lin et al. (2020), we add weight decay to local client training, effectively improving FEDAVG.

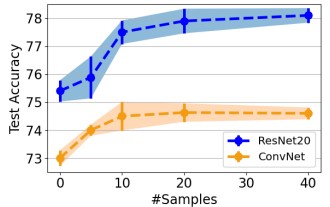

Figure 2: # of sampled models in FEDBE.

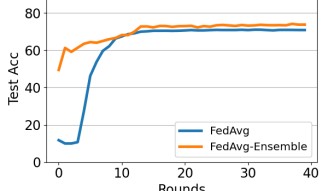

Figure 3: FEDAVG while monitoring the Bayesian ensemble.

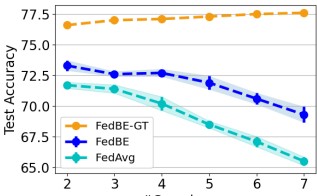

Figure 4: # of layers (ConvNet). GT: with ground-truth targets.

Table 4: FEDBE on non-i.i.d CIFAR-10 with different unlabeled data $\mathcal{U}$.

| Non-i.i.d. Type | $\mathcal{U}$ | $|\mathcal{U}|$ | ConvNet | ResNet20 | ResNet32 | ResNet44 | ResNet56 |
|---|---|---|---|---|---|---|---|
| Step | CIFAR-10 | 10K | 74.5±0.51 | 77.5±0.42 | 72.7±0.27 | 65.5±0.32 | 60.7±0.45 |
| | CIFAR-100 | 50K | 74.4±0.45 | 78.2±0.58 | 72.2±0.35 | 65.1±0.37 | 61.0±0.49 |
| | Tiny-ImageNet | 100K | 74.5±0.64 | 77.1±0.51 | 72.3±0.43 | 64.5±0.51 | 60.9±0.32 |
| Dirichlet | CIFAR-10 | 10K | 73.9±0.45 | 78.2±0.36 | 77.7±0.45 | 71.5±0.38 | 67.0±0.30 |
| | CIFAR-100 | 50K | 73.5±0.41 | 78.6±0.63 | 76.5±0.61 | 72.0±0.71 | 66.9±0.57 |
| | Tiny-ImageNet | 100K | 74.0±0.35 | 78.2±0.72 | 76.7±0.52 | 71.6±0.66 | 67.3±0.32 |

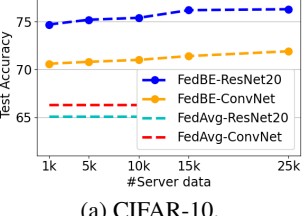

(a) CIFAR-10.

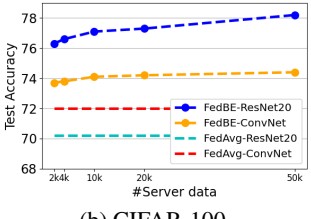

(b) CIFAR-100.

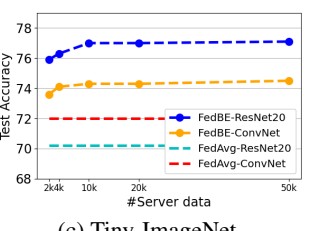

(c) Tiny-ImageNet.

Figure 5: Effects on varying the size and domains of the server dataset on CIFAR-10 experiments.

**Bayesian model ensemble vs. weight average for prediction.** In Figure 3, we perform FEDAVG and show the test accuracy at every round, together with the accuracy by Bayesian model ensemble, on the Step-non-i.i.d CIFAR-10 experiment using ResNet20. That is, we take the clients' models learned with FEDAVG to construct the distribution, sample models from it, and perform ensemble for the predictions. Bayesian model ensemble outperforms weight average at nearly all the rounds, even though it is noisy (i.e., not with $100\%$ accuracy).

**Effects of unlabeled data.** FEDBE utilizes unlabeled data $\mathcal{U}$ to enable knowledge distillation. Figure 5a studies the effect of $|\mathcal{U}|$: we redo the same Step experiments but keep 25K training images away from clients and vary $|\mathcal{U}|$ in the server. FEDBE outperforms FEDAVG even with 1K unlabeled dataset (merely $4\%$ of the total client data). We note that, FEDAVG (ResNet20) trained with the full 50K images only reaches $72.5\%$, worse than FEDBE, justifying that the gain by FEDBE is not simply from seeing more data. Adding more unlabeled data consistently but slightly improve FEDBE.

We further investigate the situation that the unlabeled data come from a different domain or task. This is to simulate the cases that (a) the server has little knowledge about clients' data and (b) the server cannot collect unlabeled data that accurately reflect the test data. In Table 4, we replace the unlabeled data to CIFAR-100 and Tiny-ImageNet. *The accuracy matches or even outperforms using CIFAR-10, suggesting that out-of-domain unlabeled data are sufficient for* FEDBE. *The results also verify that* FEDBE *uses unlabeled data mainly as a medium for distillation, not a peep at future test data.*

In Figure 5b and Figure 5c, we investigate different sizes of CIFAR-100 or Tiny-ImageNet as the unlabeled data (cf. Table 4). We found that even with merely 2K unlabeled data, which is $5\%$ of the total 40K CIFAR-10 labeled data and $2 \sim 4\%$ of the original 50K-100K unlabeled data, FEDBE can already outperform FedAvg by a margin. This finding is aligned with what we have included in Figure 5a, where we showed that a small amount of unlabeled data is sufficient for FEDBE to be effective. Adding more unlabeled data can improve the accuracy but the gain is diminishing.

**Network depth.** Unlike in centralized training that deeper models usually lead to higher accuracy (bottom row in Table 1, trained with 200 epochs), we observe an *opposite* trend in FL: all methods suffer accuracy drop when ResNets go deeper. This can be attributed to (a) local training over-fitting

Table 5: Partial participation (Tiny-ImageNet)

|  | Method | ResNet20 | MobileNetV2 |
|---|---|---|---|
| i.i.d | FEDAVG | 32.4±0.68 | 26.1±0.98 |
| | **FEDBE** | **35.4**±0.58 | **28.9**±1.15 |
| non-i.i.d | FEDAVG | 27.5±0.78 | 25.5±1.23 |
| | **FEDBE** | **32.4**±0.81 | **27.8**±0.99 |

Table 6: Systems heterogeneity (non-i.i.d. CIFAR-10)

| Method | ConvNet | ResNet20 | ResNet32 |
|---|---|---|---|
| FEDAVG | 70.6±0.46 | 69.9±0.59 | 64.0±0.50 |
| FEDPROX | 71.2±0.55 | 69.4±0.48 | 65.9±0.63 |
| **FEDBE** | 73.3±0.56 | 77.1±0.61 | 70.2±0.39 |
| **+FEDPROX** | **73.7**±0.24 | **77.5**±0.51 | **71.6**±0.37 |

to small and non-i.i.d. data or (b) local models drifting away from each other. FEDBE suffers the least among all methods, suggesting it as a promising direction to resolve the problem. To understand the current limit, we conduct a study in Figure 4 by injecting more convolutional layers into ConvNet (**Step** setting). FEDAVG again degrades rapidly, while FEDBE is more robust. If we replace Bayesian ensemble by the CIFAR-10 ground truth labels as the distillation target, FEDBE improves with more layers added, suggesting that how to distill with noisy labeled targets is the key to improve FEDBE.

## 5.3 PRACTICAL FEDERATED SYSTEMS

**Partial participation.** We examine FEDBE in a more practical environment: (a) more clients are involved, (b) each client has fewer data, and (c) not all clients participate in every round. We consider a setting with 100 clients, in which 10 clients are randomly sampled at each round and iterates for 100 rounds, similar to (McMahan et al., 2017). We study both i.i.d. and non-i.i.d (Step) cases on Tiny-ImageNet, split 10K training images to the server, and distribute the rest to the clients. For the non-i.i.d case, each client has 2 major classes (351 images each) and 198 minor classes (1 image each). FEDBE outperforms FEDAVG (see Table 5). See subsection C.7 for results on CIFAR-100.

**Systems heterogeneity.** In real-world FL, clients may have different computation resources, leading to *systems heterogeneity* (Li et al., 2020a). Specifically, some clients may not complete local training upon the time of scheduled aggregation, which might hurt the overall aggregation performance. We follow (Li et al., 2020a) to simulate the situation by assigning each client a local training epoch $E_i$, sampled uniformly from $(0, 20]$, and aggregate their partially-trained models. Table 6 summarizes the results on non-i.i.d (Step) CIFAR-10. FEDBE outperforms FEDAVG and FEDPROX (Li et al., 2020a).

## 6 DISCUSSION

**Privacy.** Federated learning offers data privacy since the server has no access to clients' data. It is worth noting that having unlabeled data *not collected from the clients* does not weaken the privacy if the server is benign, which is a general premise in the federated setting (McMahan et al., 2017). For instance, if the collected data are de-identified and the server does not intend to match them to clients, clients' privacy is preserved. In contrast, if the server is adversarial and tries to infer clients' information, federated learning can be vulnerable even without the unlabeled data: e.g., federated learning may not satisfy the requirement of differential privacy or robustness to membership attacks.

**Unlabeled data.** Our assumption that the server has data is valid in many cases: e.g., a self-driving car company may collect its own data but also collaborate with customers to improve the system. Bassily et al. (2020a;b) also showed real cases where public data is available in differential privacy.

## 7 CONCLUSION

Weight average in model aggregation is one major barrier that limits the applicability of federated learning to i.i.d. conditions and simple neural network architectures. We address the issue by using Bayesian model ensemble for model aggregation, enjoying a much robust prediction at a very low cost of collecting unlabeled data. With the proposed FEDBE, we demonstrate the applicability of federated learning to deeper networks (i.e., ResNet20) and many challenging conditions.

## ACKNOWLEDGMENTS

We are thankful for the generous support of computational resources by Ohio Supercomputer Center and AWS Cloud Credits for Research.

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

# SUPPLEMENTARY MATERIAL

We provide details omitted in the main paper.

- Appendix A: additional related work (cf. section 2 of the main paper).
- Appendix B: details of experimental setups (cf. subsection 5.1 of the main paper).
- Appendix C: additional experimental results and analysis (cf. subsection 5.2 of the main paper).
- Appendix D: additional discussions (cf. section 4 and subsection 5.2 of the main paper).
- Appendix E: additional analysis to address reviewers' comments.

## A ADDITIONAL RELATED WORK

**Federated leatning (FL).** In the multi-round federated setting, FEDAVG (McMahan et al., 2017) is the standard approach. Many works have studied its effectiveness and limitation regarding convergence (Khaled et al., 2020; Li et al., 2020b; Karimireddy et al., 2020; Li et al., 2020b; Liang et al., 2019; Stich, 2019; Zhao et al., 2018; Zhou & Cong, 2017; Haddadpour & Mahdavi, 2019), system robustness (Li et al., 2020a; Smith et al., 2017; Bonawitz et al., 2019), and communication cost (Konečnỳ et al., 2016; Reisizadeh et al., 2019), especially for the situations that clients are not i.i.d. (Li et al., 2020b; Zhao et al., 2018; Li et al., 2020a; Sahu et al., 2018) and have different data distributions, stability, etc.

**Ensemble learning and stochastic weight average.** Recent works (Huang et al., 2017; Draxler et al., 2018; Garipov et al., 2018) have developed efficient ways to obtain the base models for ensemble; e.g., by employing a dedicated learning rate schedule to sample models along a single pass of SGD training (Hsu et al., 2019). SWA (Maddox et al., 2019) applied the same learning rate schedule but simply took weight average over the base models to obtain a single strong model. We apply SWA, but for the purpose of learning with noisy labels in knowledge distillation.

**Bayesian deep learning.** Bayesian approaches (Neal, 2012; Barber, 2012; Brochu et al., 2010) incorporate uncertainty in decision making by placing a distribution over model weights and marginalizing these models to form a whole predictive distribution. Our work is inspired by (Maddox et al., 2019), which constructs the distribution by fitting the parameters to traversed models along SGD training.

**Others.** Our work is also related to semi-supervised learning (SSL) and unsupervised domain adaptation (UDA). SSL leverages unlabeled data to train a better model when limited labeled data are provided (Grandvalet & Bengio, 2005; Kingma et al., 2014; Tarvainen & Valpola, 2017; Berthelot et al., 2019; Zhu, 2005); UDA leverages unlabeled data to adapt a model trained from a source domain to a different but related target domain (Gong et al., 2012; 2014; Ganin et al., 2016; Saito et al., 2018; Ben-David et al., 2010). We also leverage unlabeled data, but for model aggregation. We note that, UDA and SLL generally assume the access to labeled (source) data, which is not the case in federated learning: the server cannot access clients' labeled data.

## B EXPERIMENTAL SETUPS

### B.1 IMPLEMENTATION DETAILS

As mentioned in the main paper (subsection 5.1), we select the number of local epochs, used in training a client model within one round of communication, according to the performance of FEDAVG. Other algorithms, like FEDBE and v-Distillation, then follow the same numbers. For ConvNet and ResNet experiments, we fixed $E = 20$. We use $E = 10$ for MobileNet experiments. For all 1-Ensemble baselines, we tuned $E$ from $[10, 20, ..., 200]$.

We observed that applying weight decay in local client training improves all FL methods, but the suitable hyper-parameter can be different for different methods on different network architectures. Tuning it specifically for each method is thus essential for fair comparisons. We search the weight decay hyper-parameter for each network and each method in $[1e-3, 1e-4]$ with a validation set.

For methods with distillation (FEDBE and v-Distillation), We tune the epochs for v-Distillation from $[1, 5, 10, 20, 30, 40]$ and find 20 to be stable across different setups. We apply 20 to FEDBE as well.

In constructing the pseudo-labeled data $\mathcal{T}$, we perform inference on the unlabeled data $\mathcal{U}$ in the server without data augmentation. We perform data augmentation in both local training on $\mathcal{D}_i$ and knowledge distillation on $\mathcal{T}$. The $32 \times 32$ CIFAR images are padded 2 pixels each side, randomly flipped horizontally, and then randomly cropped back to $32 \times 32$. The $64 \times 64$ Tiny-ImageNet images are padded 4 pixels each side, randomly flipped horizontally, and then randomly cropped back to $64 \times 64$. In Table 4 of the main paper, we resize images of Tiny-ImageNet to $32 \times 32$.

For neural networks that contain batch normalization layers (Ioffe & Szegedy, 2015), we apply the same way as in section 3 to construct the global distribution for the layers and we observe no issues in our experiments. SWAG (Maddox et al., 2019) also reported that it performs stably even on very deep networks.

## B.2 TRAINING FEDAVG

For the local learning rate $\eta_l$, we observed that an appropriate value of $\eta_l$ is important when training on the non-i.i.d local data. The local models cannot converge if the learning rate is set to a too large value (also shown in (Reddi et al., 2020)), and the models cannot reach satisfying performance within the local epochs $E$ with a too-small value as shown in Figure 6. Also, unlike the common practice of training a neural network with learning rate decay in the centralized setting, we observed that applying it within each round of local training (decay by 0.99 every step) results in much worse client models. The local models would need a large enough learning rate to converge with a fixed $E$ and we use 0.01 as the base learning rate for local training in all our experiments.

Although decaying $\eta_l$ within each round of local training is not helpful, decaying $\eta_l$ along the rounds of communication could improve the performance. Wang et al. (2020) and Reddi et al. (2020) provided both theoretical and empirical studies suggesting that the local learning rate must decay along the communication rounds to alleviate client shifts in non-i.i.d setting. In our experiments, at the $r$th round of communication, the local client training starts with a learning rate $\eta_l$, which is 0.01 if $r < 0.3R$, 0.001 if $0.3R \leq r < 0.6R$, and 0.0001 otherwise, where $R$ is the total rounds of communication. In Figure 7, we examined this schedule with different degrees of $\alpha$ in the Dirichlet-non-i.i.d setting. We observed consistent improvements and applied it to all our experiments in section 5.

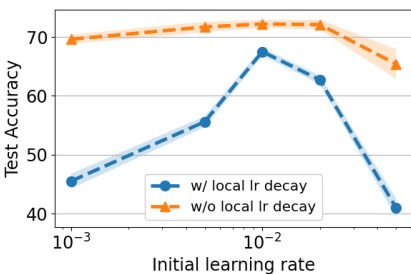
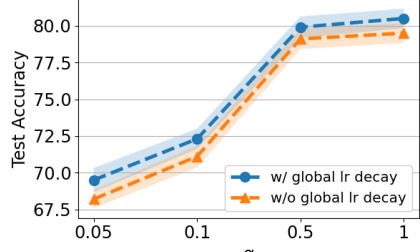

Figure 6: FEDAVG with ConvNet on Step-non-i.i.d CIFAR-10 with or without learning rate decay within each round of local training.

Figure 7: FEDAVG with ConvNet on Dirichlet-non-i.i.d CIFAR-10 with or without learning rate decay at latter rounds of communication. We experimented with different values of $\alpha$ in Dir($\alpha$).

## B.3 EFFECTS OF WEIGHT DECAY IN LOCAL CLIENT TRAINING

Federated learning on non-i.i.d data of clients is prone to model drift due to the deviation of local data distributions and is sensitive to the number of local epochs $E$ (Wang et al., 2020; Li et al., 2020b; McMahan et al., 2017). To prevent local training from over-fitting the local distribution, we apply $\ell_2$ regularization as weight decay. In a different context of distributed deep learning, Sagawa et al. (2020) also showed that $\ell_2$ regularization can improve the generalization by preventing the local model from perfectly fitting the non-i.i.d training data.

Table 7: FEDBE with models sampled from a Dirichlet distribution on Step-non-i.i.d. CIFAR-10. We compare different $\alpha \times \mathbf{1}$ in setting the parameter of a Dirichlet distribution.

| $\alpha$ | ConvNet | ResNet20 |
|---|---|---|
| 0.1 | 72.5±0.44 | 75.9±0.66 |
| 0.5 | 73.6±0.73 | 77.3±0.86 |
| 1 | 74.2±0.51 | 77.1±0.71 |
| 2 | 73.2±0.83 | 76.8±0.55 |

As shown in Figure 8 where we compared FEDAVG and FEDBE with or without weight decay in local training, we found that weight decay not only leads to a higher test accuracy but also makes both algorithms more robust to the choice of local epochs $E$.

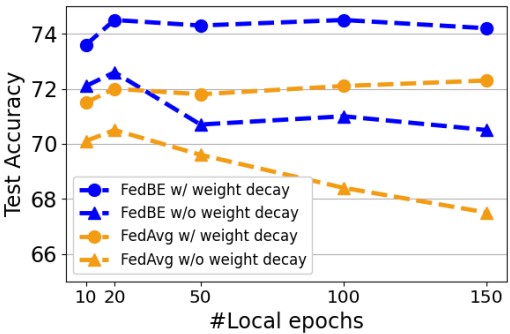

Figure 8: FEDAVG and FEDBE with ConvNet on CIFAR-10 (Step-non-i.i.d) with or without weight decay in local training, for different numbers of local epochs $E$.

## C  EXPERIMENTAL RESULTS AND ANALYSIS

### C.1  GLOBAL MODEL SAMPLING IN FEDBE

In the main paper, we mainly report accuracy by FEDBE with models sampled from a Gaussian distribution (cf. Equation 5 of the main paper). Here we report results using a Dirichlet distribution for model sampling (cf. Equation 7 of the main paper) in Table 7. We compare different $\boldsymbol{\alpha} = \alpha \times \mathbf{1}$ in setting the parameter of a Dirichlet distribution. We see that the accuracy is not sensitive to the change of $\alpha$. Compared to Table 1 of the main paper, FEDBE with Dirichlet is slightly worse than FEDBE with Gaussian (by $\leq 0.5\%$) but much better than FEDAVG.

To further study the models sampled from the global model distribution constructed in FEDBE, we compare the prediction accuracy and confidences (the maximum values of the predicted probabilities) of clients' models and sampled models. We show the histogram of correctly and incorrectly predicted test examples at different prediction confidences. As shown in Figure 9, we observe that clients' models tend to be over-confident by assigning high confidences to wrong predictions. We hypothesize that it is because clients' local training data are scarce and class-imbalanced. On the other hand, sampled models have much better alignment between confidences and accuracy (i.e., higher confidences, higher accuracy).

### C.2  ANALYSIS ON WEIGHT AVERAGE, (BAYESIAN) MODEL ENSEMBLE, AND DISTILLATION

To investigate the difference of weight average, (Bayesian) model ensemble, and distillation in making predictions, we focus on one-round federated learning, in which the client models are trained in the same way regardless of what aggregation approach to be used. We experiment with Step-non-i.i.d. CIFAR-10 using ConvNet, and train the 10 local client models for 200 epochs. We then compare (a) weight average to combine the models, (b) model ensemble, and (c) Bayesian model ensemble

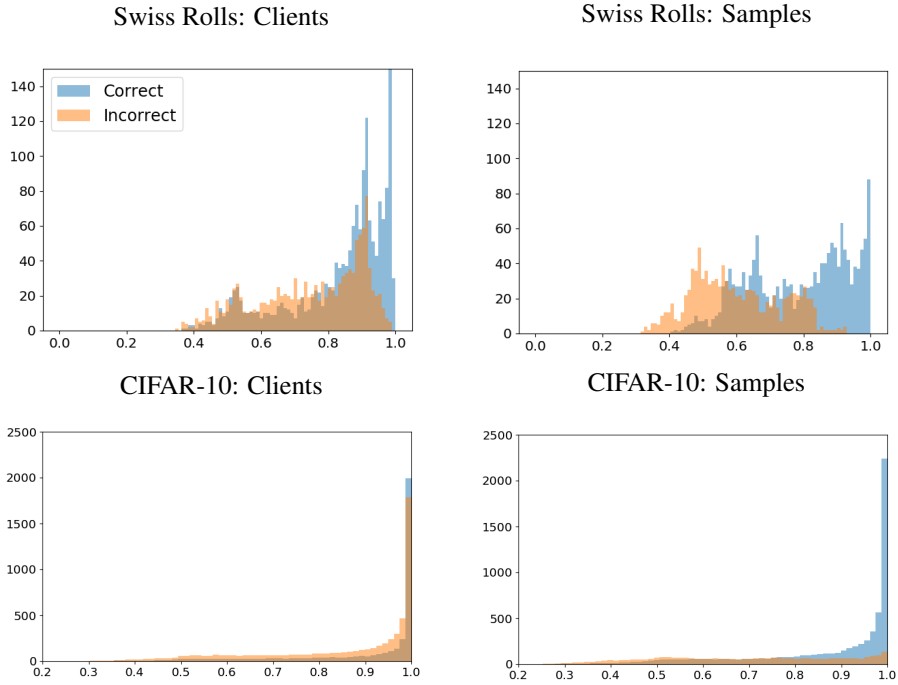

Figure 9: Histograms of correctly and incorrectly predicted examples (vertical axes) along the confidence values (the maximum values of the predicted probabilities). Upper row: Swiss roll dataset used in Figure 1 of main paper (averaged over 3 clients or sampled models); lower row: Step-non-i.i.d. CIFAR-10 (averaged over 10 clients or sampled models).

Table 8: One-round federated learning on Step-non-i.i.d. CIFAR-10 with ConvNet. We compare different strategies to combine the clients' local models, including weight average, (Bayesian) model ensemble, and ensemble distillation (with SGD or SWA).

| Method | Distillation | | |
|---|---|---|---|
| | None | SGD | SWA |
| (a) Weight average | 24.7±0.85 | - | - |
| (b) Model ensemble | 60.5±0.28 | 32.0±0.74 | 33.1±1.02 |
| (c) Bayesian model ensemble | 62.5±0.35 | 35.1±0.76 | 35.7±0.86 |

with $M = 10$ extra samples beyond weight average and individual clients. For (b) and (c), we further apply knowledge distillation using the unlabeled server data to summarize the ensemble predictions into a single global model using SGD or SWA. We note that, method (a) is equivalent to one-round FEDAVG; method (b) without distillation is the same as 1-Ensemble; method (b) with SGD distillation is equivalent to one-round v-Distillation; method (c) with SWA is equivalent to one-round FEDBE. Table 8 shows the results with several interesting findings. First, without distillation, model ensemble clearly outperforms weight average and Bayesian model ensemble further adds a 2% gain, supporting our claims in section 3. Second, summarizing model ensemble into one global model largely degrades the accuracy, showing the challenges of applying ensemble distillation in federated learning. The 60.5% and 62.5% accuracy by ensemble, although relatively higher than others, may still be far from perfect to be used as distillation targets. Third, distillation with SWA outperforms SGD for both model ensemble and Bayesian model ensemble, justifying our proposed usage of SWA. Fourth, with or without distillation, Bayesian model ensemble always outperforms model ensemble, with very little cost of estimating the global model distribution and performing sampling. Fifth, even with the degraded accuracy after distillation, model ensemble and Bayesian model ensemble, after distilled into a single model, still outperforms weight average notably. Finally, although in the one-round setting we hardly see the advantage of distilling the model ensemble into a single model,

FEDAVG: inference on CIFAR-10    FEDAVG: inference on CIFAR-100

FEDBE: inference on CIFAR-10    FEDBE: inference on CIFAR-100

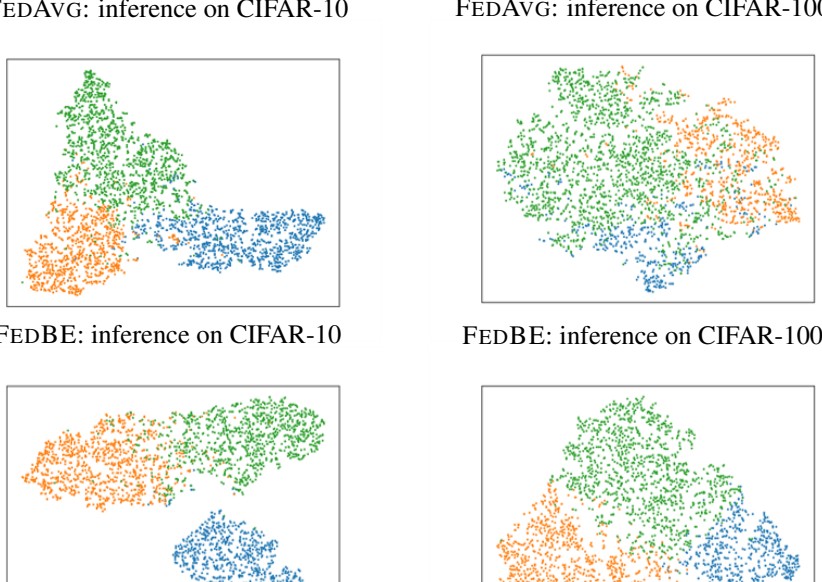

Figure 10: Feature visualization of FEDAVG and FEDBE models. All models are trained on Step-non-i.i.d. CIFAR-10, then inference on CIFAR-10/100 test sets. The features are colored with the ground-truth labels (CIFAR-10) or the predictions (CIFAR-100).

with multiple rounds of communication as in the main paper (cf. subsection 5.2), its advantage becomes much clear—it allows the next-round local training to start from a better initialization (in comparison to weight average) and eventually leads to much higher accuracy than 1-Ensemble.

## C.3    FEDBE vs. FEDAVG

We provide further comparisons between FEDBE and FEDAVG. First, we perform Bayesian model ensemble at the end of FEDAVG training (Step-non-i.i.d. CIFAR-10). We achieve 72.5% with ResNet20, better than FEDAVG (70.2% in Table 1 of the main paper) but still worse than FEDBE (77.5%), demonstrating the importance of incorporating Bayesian model ensemble into multi-round federated learning.

To further analyze why FEDBE improves over FEDAVG, we train models on Step-non-i.i.d CIFAR-10, inference on the test sets, and visualize the features. We trained a FEDBE ResNet20 model for only 15 rounds such that the test accuracy is similar to a FEDAVG ResNet20 model trained for 40 rounds. In Figure 10, we plot their features using t-SNE (Maaten & Hinton, 2008) on the CIFAR-10/100 testing sets (consider 3 semantically different classes: automobile, cat, and frog) and color the features with the ground-truth labels of CIFAR-10 test set or the predictions of the models on CIFAR-100 test set. Interestingly, we observe that the features of FEDBE are more discriminative (separated) than the features of FEDAVG, especially on CIFAR-100, even if FEDBE and FEDAVG have similar test accuracy on CIFAR-10.

We further discuss Table 3 of the main paper. We perform data augmentation on $\mathcal{T}$ in knowledge distillation. This explains why we obtain improvement over FEDAVG when using the FEDAVG predictions as the target labels (72.6%/73.4% vs. 72.0%/70.2% in Table 1 of the main text, using ConvNet/ResNet20). We note that without data augmentation, using FEDAVG predictions as the target leads to zero gradients in knowledge distillation since we initialize the student model with FEDAVG. The results suggest the slight benefit of collecting unlabeled data for knowledge distillation in model aggregation.

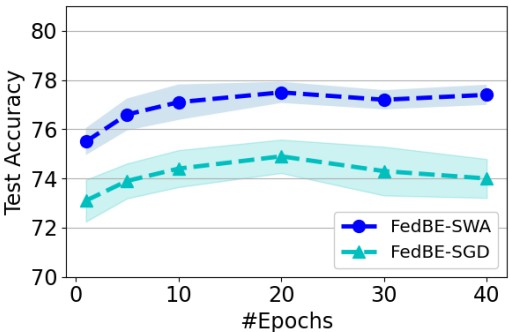

Figure 11: ResNet20 test accuracy on Step-non-i.i.d. CIFAR-10, with different numbers of epochs for distillation using SGD and SWA for FEDBE.

## C.4    FEDBE WITH SWA AND SGD

We found that distillation with SGD is more sensitive to noisy labels and the number of epochs. For ResNet20 (Table 1), FEDBE with FEDAVG + C + S w/o SWA (i.e., using SGD) achieves 74.9% with 20 epochs but 74.0% with 40 epoch. In contrast, FEDBE with SWA is much stable. As shown in Figure 11, the accuracy stays stable with more than 10 epochs being used, achieving 77.5% with 20 epochs and 77.3% with 40 epochs.

## C.5    BATCH NORMALIZATION VS. GROUP NORMALIZATION

Hsieh et al. (2020) showed that FEDAVG in non-i.i.d cases can be improved by replacing the batch normalization (BN) layers with group normalization (GN) layers (Wu & He, 2018). However, we observe that ResNets with GN converge much slower, which is consistent with the observations in (Zhang et al., 2020). In our CIFAR-10 (Step) experiments, FEDAVG using ResNet20 with GN can outperform that with BN slightly if both are trained with 200 rounds (76.4% vs. 74.6%). FEDBE can further improve the performance: FEDBE with GN/BN achieves 79.6%/80.2%.

## C.6    COMPATIBILITY WITH SCAFFOLD

SCAFFOLD (Karimireddy et al., 2020) is a recently proposed FL method to regularize local training. We experiment with SCAFFOLD on non-i.i.d. (Step) CIFAR-10. We find that SCAFFOLD cannot directly perform well with deeper networks (ResNet20: 59.4%; ResNet32: 55.3%). Nevertheless, FEDBE +SCAFFOLD can improve upon it, achieving 76.4% and 72.7%, respectively.

To further analyze why FEDBE improves SCAFFOLD, we plot the test accuracy of SCAFFOLD vs. FEDBE +SCAFFOLD at every communication round. We see that both methods perform similarly in the early rounds. SCAFFOLD with weight average could not improve the accuracy after roughly 10 rounds. FEDBE +SCAFFOLD, in contrast, performs Bayesian ensemble and distillation to obtain the global model, bypassing weight average and gradually improving the test accuracy. We therefore argue that, as long as FEDBE can improve SCAFFOLD slightly at every later round, the ultimate gain can be large. We also attribute the gain brought by FEDBE to the robustness of Bayesian ensemble for model aggregation.

## C.7    PARTIAL PARTICIPATION ON CIFAR-100 (CF. SUBSECTION 5.3)

We also conduct the experiments on CIFAR-100 with the non-i.i.d. **Step** setting. We consider a setting with 100 clients, in which 10 clients are randomly sampled at each round and iterates for 100 rounds, similar to (McMahan et al., 2017). We split 10K images from the 50K training images to the server, and distribute the remaining ones to the clients. Each client has 5 major classes (61 images each) and 95 minor classes (1 image each). Table 9 shows the results: FEDBE consistently outperforms FEDAVG.

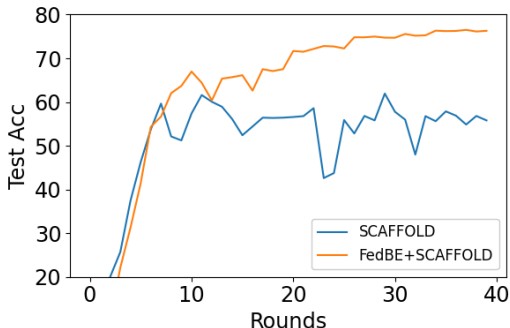

Figure 12: Step-non-i.i.d CIFAR-10 experiments accuracy curves of SCAFFOLD on ResNet20.

Table 9: Non-i.i.d CIFAR-100

| Method | ConvNet | ResNet20 | ResNet32 |
|--------|---------|----------|----------|
| FEDAVG | 32.5±0.78 | 37.5±0.65 | 33.3±0.55 |
| **FEDBE** | **36.6**±0.52 | **43.5**±0.89 | **37.7**±0.69 |

# D DISCUSSION

## D.1 EXTRA COMPUTATION COST

FEDBE involves more computation compared to FEDAVG. The extra cost is on the server and no extra burden is on the clients. In practice, the server is assumed to be computationally rich so the extra training time is negligible w.r.t. communication time. Using a 2080 Ti GPU on CIFAR-10 (ConvNet), building distributions and sampling takes 0.2s, inference of a model takes 2.4s, and distillation takes 10.4s. Constructing the ensemble predictions $\mathcal{T} = \{(\boldsymbol{x}_j, \hat{\boldsymbol{p}}_j)\}_{j=1}^{J}$, where $\hat{\boldsymbol{p}}_j = \frac{1}{M} \sum_{m=1}^{M} p(y|\boldsymbol{x}_j; \boldsymbol{w}^{(m)})$, requires each $\boldsymbol{w}^{(m)}$ to be evaluated on $\mathcal{U}$, which can be easily parallelized in modern GPU machines. The convergence speed of the Monte Carlo approximation in Equation 4 is $1/\sqrt{M}$, yet we observe that $M = 10 \sim 20$ is sufficient for Bayesian model ensemble to be effective.

## D.2 FEDAVG ON DEEPER NETWORKS

Deeper models are known to be poorly calibrated (Guo et al., 2017), especially when trained on limited and imbalanced data. The loss surfaces can be non-convex (Garipov et al., 2018; Draxler et al., 2018). FEDAVG thus may not fuse clients well and may need significantly more rounds of communication with local training of small step sizes to prevent client's model drifting.

# E FURTHER ANALYSIS

## E.1 TEST ACCURACY AT DIFFERENT ROUNDS

We follow the experimental setup in section 5 and further show the test accuracy of the compared methods at different communication rounds (in total 40 rounds) in Figure 13. Specifically, we experiment with ResNet20 and ResNet32 for both the Step-non-i.i.d. and Dirichlet-non-i.i.d. settings on CIFAR-10 (the results at 40 rounds are the same as those listed in Table 1). FEDBE obtains the highest accuracy after roughly 10 rounds, and could gradually improve as more rounds are involved. Interestingly, v-Distillation, which performs ensemble directly over clients' models without other sampled models, normally obtains the highest accuracy in the first 10 rounds, but is surpassed by FEDBE after that. We hypothesize that in the first 10 rounds, as the clients models are still not well trained, the constructed distributions may not be stable. We also note that except 1-Ensemble,

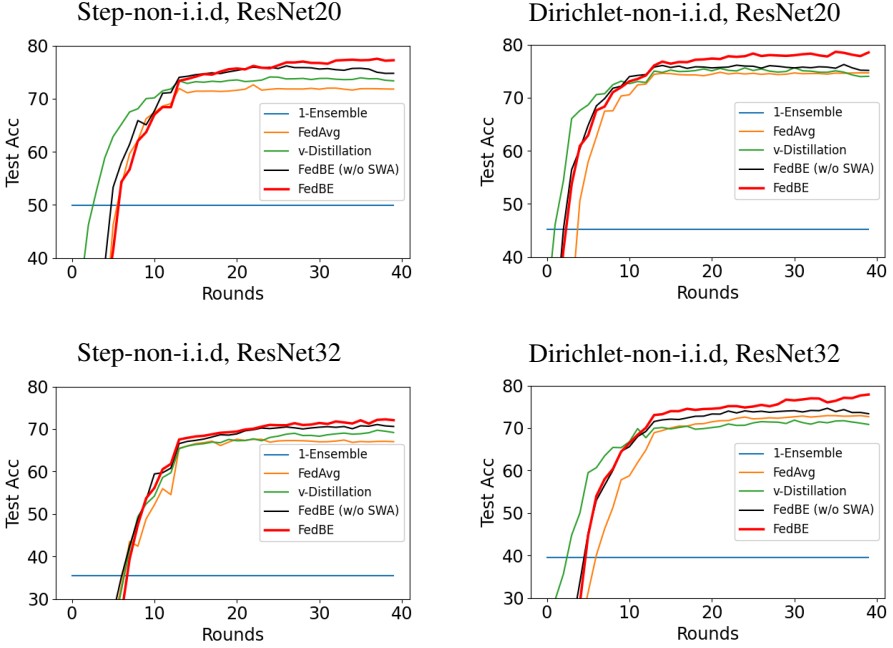

Figure 13: CIFAR-10 curves of test accuracy at different communication rounds. We study both non-i.i.d settings (Step and Dirichlet) using ResNet20 and ResNet32 (cf. subsection E.1).

the other methods with ensemble mostly outperform FEDAVG in the first 10 rounds, showing their robustness in aggregation.

