# OpenReview forum: "FedBE: Making Bayesian Model Ensemble Applicable to Federated Learning"
_ICLR.cc/2021/Conference — ICLR 2021 Poster_

### Official Review · AnonReviewer1 · 2020-10-28
**Interesting idea but some experimental results need further clarification**

**Rating:** 6
**Confidence:** 4

**Review:**

Summary:

The paper proposes a new model aggregation approach named FedBE in federated learning. In each round, FedBE first fits the local models into a Gaussian/Dirichlet distribution as a posterior distribution of the global model. Then, FedBE uses the sampled models from the distribution, the local models, and the averaged model as an ensemble to label a public dataset. The labeled dataset is used to train a global model. The experiments show that FedBE can outperform FedAVG on CIFAR-10 using ConvNet and ResNet.

Pros:

(1) The paper is well written and easy to follow.

(2) The studied problem is important and the proposed idea is interesting.

(3) The experiments are comprehensive and cover many aspects.

I have the following concerns, mainly about the experiments.

(1) In Algorithm 1, only the sampled models are used to predict the unlabeled dataset. However, in the experiments, the sampled models, the local models, and the averaged model are all used as an ensemble. Please keep consistency. Also, if the local models may perform badly due to client’s model drift, why the local models are also included in the ensemble?

(2) In Section D.6, SCAFFOLD has a bad test accuracy. I don’t understand why FedBE can improve SCAFFOLD a lot. If the averaged model is bad in SCAFFOLD, the sampled models are also very likely to be bad. However, the authors show that the accuracy of FedBE+SCAFFOLD is still close to FedBE+FedAvg. Can the authors explain it?

(3) From Table 7, FedBE with Dirichlet distribution is worse than FedBE with Gaussian distribution although it introduces an additional parameter $\alpha$. Is there any explanation?

(4) The size of unlabeled public data is large in the experiments. The paper uses CIFAR100+tiny-imagenet as the public dataset, which has a much larger size than the training dataset (i.e., CIFAR10). However, in reality, the data are usually private and the server may not be able to collect a large related public dataset. Can the authors show the performance of FedBE with a smaller public dataset?

(5) The authors can add a figure to show the test accuracy of different approaches versus the number of communication rounds. It is more clear to see the performance of different approaches with such a figure.

(6) The paper presents a simple averaging version in the main text and a weighted averaging version in Appendix. Which version is used in the experiments?

Minor issue: Figure 1 uses Dirichlet distribution as an example. Since the paper focuses on the Gaussian distribution, the authors can present an example based on Gaussian distribution instead.


==========Post rebuttal============

Thanks for the authors' response. The authors have addressed some of my concerns. I have raised my score accordingly.

---

> ### Author Response · Authors · 2020-11-22
> **Thanks for your comments!**
>
> Thank you for the constructive comments.
>
> **Algorithm 1:** Thanks and we have updated it accordingly. We have also provided a detailed comparison of what models are included in the ensemble (please see Table 3): it is worth noting that both the local models and weight average are possible samples from the distribution. Overall, including local clients in the ensemble reaches better performance. This is not surprising for two reasons. First, ensemble methods require “diverse” base models (Zhou, 2012; Dietterich, 2000; Breiman, 1996; Kuncheva and Whitaker, 2003). While each of the client models does not perform well on the test data alone, they are trained to converge to the local data and can be treated as local experts, similar to what each decision tree (in a random forest) does to subsampled data. Figure 1 (c) and Figure 11 show that the client models are different from many of the sampled models, showing that they increase the diversity of the base models. Second, ensemble methods are known to combine weak base models into a stronger one. While each of the client models may drift away from others, their direct ensemble is indeed robust, suggesting them as qualified base models. This can be seen in section D.2 and Table 8. In Table 8, we show one-round federated learning: a direct ensemble over clients model significantly outperforms the weight average, and adding sampled models further boosts the performance.
>
> (Kuncheva and Whitaker) Measures of diversity in classifier ensembles and their relationship with the ensemble accuracy. Machine learning, 2003
>
> **FedBE and SCAFFOLD:** We note that the sampled models are likely bad does not mean that the ensemble will be bad. This can be seen again from section D.2 and Table 8. We therefore attribute the gain brought by FedBE to the robustness of Bayesian ensemble for model aggregation. We also note that both FedBE and SCAFFOLD are multi-round algorithms. In other words, as long as FedBE can improve SCAFFOLD slightly at every round, the ultimate performance can be quite different. This can be seen from Figure 14 (and section D.6), in which we provide the test accuracy at each round. FedBE and SCAFFOLD perform similarly in the early rounds. SCAFFOLD with weight average could not improve the accuracy after certain rounds. FedBE + SCAFFOLD, in contrast, performs Bayesian ensemble and distillation to obtain the global model, bypassing weight average and gradually improving the test accuracy.
>
> **Dirichlet v.s. Gaussian:** This is a good question, and we attribute this to that Gaussian can lead to more diverse samples than Dirichlet. Sampling from Dirichlet is essentially equivalent to performing random convex combinations of clients model, while sampling from Gaussian could go beyond convex combinations. In fact, Gaussians have been commonly used to approximate the posterior of model weights (Blundell et al., 2015; Kirkpatrick et al., 2017; Lee et al., 2018). We note that, having the parameter “\alpha” does not mean that Dirichlet is more powerful or has more parameters than Gaussian. Gaussian estimates the mean and variance. The way we use Dirichlet in Eq. 7 is to sample coefficients to combine client models, more like a non-parametric method of model distributions.
>
> **Size of unlabeled data:** Thanks for the suggestion and we have included them in Figure 5 and Figure 6. We investigate different sizes of CIFAR-100 or Tiny-ImageNet as the unlabeled data. We found that even with merely 2K unlabeled data, which is 5% of the total 40K CIFAR-10 labeled data and 2-4% of the original 50K-100K unlabeled data, FedBE can already outperform FedAvg by a margin. This finding is aligned with what we have included in Figure 3, where we showed that a small amount of unlabeled data is sufficient for FedBE to be effective. Adding more unlabeled data can improve the accuracy but the gain is diminishing.
>
> We note that CIFAR-100 or Tiny-ImageNet contains images from different domains or tasks (i.e., different categories) and are already not that related to CIFAR-10. In some applications, it can be feasible for the server to collect a large number of these unlabeled data. For instance, by crawling from the Internet.
>
> **Accuracy over rounds:** Thank you for the suggestion and we provide such figures in Figure 15 (section F.1).
>
> **Which version?** Throughout all our experiments, we used the correct, weighted version described in the Appendix. We presented the simple average version in the main text to ease the notation. Following reviewer 3’s suggestion, we have incorporated the weighted version into the main text.
>
> **Gaussian in Figure 1:** Thanks for the suggestion. We note that, however, drawing such a Gaussian is difficult since the distribution is in a high-dimensional space and can not be trivially visualized in 2D (like the one in Figure 1 (b)). This is why we plotted Dirichlet, whose samples are convex combinations of the 3 client models and can be placed in a 2-simplex.

---

### Official Review · AnonReviewer2 · 2020-10-28
**Questions about using unlabeled data**

**Rating:** 5
**Confidence:** 2

**Review:**

This paper focuses on improving model aggregation on the server of multi-round federated learning. For this purpose, this paper proposes FedBE, which combines the idea of Bayesian model ensembling and distillation based on stochastic weight averaging. A set of unlabelled data on the server is required for distillation. Experiments are conducted on CIFAR-10/100 and tiny-ImageNet with both IID and non-IID cases.

The paper is well written and easy to follow. Though Bayesian model ensemble and distillation based on SWA are not new topics, using the idea of them in the field of federated learning is novel to the best of my knowledge.

My major concern is about the experiment setting for using unlabeled data. According to my understanding, from the main body of the paper (Sec 5.2), 10K training images are used as unlabeled data saved on the server and the rest is distributed to each client. For baseline method like FedAvg, these 10K images are useless since it does not need unlabeled data for model average on the server. The different number of total training instances between FedBE (40K labeled + 10K unlabeled) and FedAvg (40K labeled) makes the comparison unfair.

Besides, the analysis about either computational complexity or empirical wall clock time is needed. FedBE replaces the original model average with ensemble prediction & training, which is very slow. And these computations on the server seem cannot be parallelized with the computation on the client.  As a result, the computation on the server maybe the bottleneck of FedBE. So providing detailed analysis is important for better evaluating FedBE.

---

> ### Author Response · Authors · 2020-11-22
> **Thanks for your comments!**
>
> We thank the reviewer for the constructive comments. We address them as follows:
>
> **Fair comparisons using unlabeled data:** This is a good question, and we did conduct an experiment where we applied FedAvg using all the 50K data (40K labeled + 10K labeled) in the original submission.  Please be referred to section 5.2 “Effects of unlabeled data”. FedAvg with 50K labeled data achieves 72.5% (ResNet20), which is still worse than 77.5% by FedBE (40K labeled + 10K unlabeled).
>
> Here we provide some more results. We provide the full row of FedAvg trained with all 50K labeled CIFAR-10 training images on Step-non-i.i.d setting (cf. Table 1):
>
> Method: ConvNet / ResNet20 / ResNet32 / ResNet44 / ResNet56
> FedAvg (50K labeled): 72.6 /  72.5 / 68.1 / 61.5 / 56.4
> FedBE (40K labeled + 10K unlabeled): 74.5 / 77.5 / 72.7 / 65.5 / 60.7
>
> FedBE still consistently outperforms FedAvg.
>
> We also note that, even with a smaller number of unlabeled data or with unlabeled data from other domains/tasks (e.g., CIFAR-100, Tiny-ImageNet), FedBE can already outperform FedAvg. (Please be referred to the paragraph “Effects of unlabeled data” in section 5.2 and Figure 3 and Table 4.)
>
> In practice, we think comparing FedBE to FedAvg with the same number of “labeled” data is still more meaningful. First, the labeled data are from the clients (so both methods have the same number of them), while the unlabeled data are collected by the server. Second, the unlabeled data may not be from the same task or domain as the labeled data.  Both situations make combining the server data with the clients’ data difficult for FedAvg.
>
> **Computation cost:** Please see the general response.

---

### Official Review · AnonReviewer4 · 2020-10-29
**Nice combination of different techniques to construct a good algorithm with extensive evaluation**

**Rating:** 7
**Confidence:** 3

**Review:**

The paper presents FEDBE, a novel method for federated learning. Models are trained locally in the clients' side then communicated to the server. The global model is constructed by Bayesian model ensemble. Knowledge distillation is used to construct the model that is passed to the clients for the next round. FEDBE is simple and shows strong empirical performance.

Strong points:

- The method demonstrates strong performance versus common baselines
- Extensive experiments were conducted.
- The paper is well written.


Clearly state your recommendation (accept or reject) with one or two key reasons for this choice.
- I recommend to accept this paper.
- This type of work is definitely needed to enable more powerful models that preserves the privacy of the clients.


Questions:

- What is the effect of number of epochs between model collections (25 steps vs. 50 or more)? Is this what is shown in Figure 10? If so please clarify more.
- What is the effect of the step size decaying on SWA? IS the method sensitive to this choice?
- What are the hyperparameters that the method is sensitive for?

---

> ### Author Response · Authors · 2020-11-22
> **Thanks for your comments!**
>
> We thank the reviewer for the positive review and constructive comments. We address them as follows:
>
>
> **Hyperparameters of SWA:** The number of “steps” (or iterations) between model collections is a hyperparameter of SWA. We apply SWA to distillation at the server, at every round of federated learning (FL). SWA employs a cyclical learning rate schedule every Q steps (i.e., decaying the learning rate within Q steps) and collects the model at the end. The learning rate is then reset to start the next cycle. These cycles allow the models to escape from local minimums; the learning rate decay allows the model to converge to a local minimum. The period Q of the cycle controls how many steps we want to optimize for each local minimum and how many models we can collect. We set Q = 25 steps and learning rate decay from 10^-3 to 4*10^-4. We have also tried other values (e.g., Q from 10 steps to 100 steps), but we did not see much performance difference.
> Figure 13 (originally Figure 10) shows the number of “epochs” for distillation at each FL round, which is not the steps between model collections in SWA.
>
> **Other hyperparameters specific to FedBE are** # of sampled models (Figure 2), # of unlabeled data at the server (Figure 3), base models included in the ensemble (Table 3), and the number of epochs for distillation on the server (Figure 13). We have provided the ablation study for each of them. For the # of sampled models and # of unlabeled data, the more the better, but we see diminish gains. For base models included in the ensemble, using all three types of base models is usually the best. For the number of “epochs” for distillation, we found a number larger than 10 is quite stable. A number smaller than 10 leads to degradation.

---

### Official Review · AnonReviewer3 · 2020-11-06
**A bit limited novelty, but an interesting contribution**

**Rating:** 6
**Confidence:** 4

**Review:**

-- Summary --
The paper proposes a new approach to model aggregation at each round performed on the server using a combination of Bayesian ensembling, model distillation, and stochastic weight averaging. The proposed method is essentially a drop-in replacement for weighted averaging based model aggregation used by FedAvg. The new method, FedBE, is extensively evaluated experimentally on federated image classification datasets where the authors show improved performance over multiple baselines.


-- Overall evaluation --
I find the paper well written and the idea of using ensembles and distillation for model aggregation in FL settings very interesting. Although the overall idea is identical to FedDF by Lin et al. (2020) who proposed to use ensembling and distillation and the authors cite and compare with that work, I think forming the ensemble distribution, sampling from it, as well as using SWA for distillation are all novel components; the contribution of each introduced component is empirically analyzed in ablation studies. Overall, while I don't find the approach extremely novel or insightful, it seems like a nice contribution to the literature. Below are some comments and questions that I hope will help further clarify and improve the paper.


-- Comments and questions --

- Section 3.3: While I see how different distributions over w are formed (Gaussian or Dirichlet), I'm not sure I understand why these distributions approximate the actual posterior? In other words, is it true that the distributions described by the authors minimize KL (or some other) divergence to the true posterior? Can you justify that?

- Regarding the Dirichlet (eq. 7), how are the gamma parameters estimated? Or are these simply hyperparameters?

- It's nice to have a toy example in section 3 (Figure 1) that illustrates the approach. However, some details are missing from the description. For example, how alphas were selected? How many clients per round were used? How did the test data distribution differ from the data on the clients?

- Computational complexity: I wonder how much is the computational overhead one has to pay to run FedBE on the server instead of FedAvg? In some practical settings, FedBE could be a great replacement for FedAvg, but in other settings, it might be too compute-intensive. It would be nice if the authors can discuss tradeoffs and analyze computational complexity.

- Related to the previous question: I wonder if distillation at each round is actually necessary? Given an ensemble (which is relatively cheap to obtain), why not sample multiple w's from it and initialize local models on clients using these samples? Do the authors expect it to work worse?


-- Experiments --

- Table 1 clearly demonstrates that FedBE dominates other methods on CIFAR-10. However, I'm puzzled by the reverse trends for centralized SGD and FL with the increase in the complexity of the model: more complex models trained with FL have worse performance than simpler models, and the opposite is true for centralized training. I wonder if ResNets were actually undertrained to some extent? Can the authors explain this phenomenon?

- Section 5.1, paragraph 2: how come the number of local epochs is computed as E = K / batch_size? Also, which metric was used exactly to tune the number of local epochs for FedAvg? After how many rounds the performance of FedAvg was evaluated and used for selecting E?


-- Minor --

- Is there any empirical evidence that supports the claim in footnote 2?

- Eq. 2 assumes that all clients have local datasets of the same size. Appendix A (eq. 10) provides the correct expression, but why not give that expression directly in the main text? The difference is literally 2 symbols. Same for eqs. 11-13.

---

> ### Author Response · Authors · 2020-11-22
> **Thanks for your comments!**
>
> Thank you for the constructive comments.
>
> **Dirichlet (eq. 7):** \gamma^{(m)} is the outcome sampled from the Dirichlet. \alpha is the parameter of the Dirichlet and we treat it as a hyperparameter. In Table 7, we show the accuracy with different \alpha: \alpha = 0.5 or 1.0 is stable.
>
> **Toy example:** We have provided some details in the original submission (section 3.3, Analysis) and will further clarify it. We set \alpha = 0.5 since it is stable (as mentioned above). There are three clients and all of them are used at every round. The test data has three “balanced” classes whose data are distributed along Swiss Rolls. The three clients’ data are distributed along the same Swiss Rolls, but the class proportions are “not balanced”. Each client has 80% of its data from one class and 20% from the other two.
>
> **Computation:** Please see the general response.
>
> **Is distillation necessary?** We aim to improve model aggregation, sending high-quality models back to clients for local training. While the Bayesian ensemble performs well, it does not mean every sampled base model performs well. For example, in Figure 1 and section 3.3 Analysis, each sampled model is not necessarily better than the weight average used in FedAvg, but their ensemble outperforms the weight average. (See also Figure 7.) This is why we perform distillation, to distill the Bayesian model ensemble into a single model that can be sent back to clients.
>
> To further justify this, we conducted two experiments with ResNet20 on Step-non-i.i.d CIFAR-10, on which FedBE gets 77.5% accuracy in Table 1.
>
> Exp1: We implement the idea the reviewer suggested. At every round, we sample 10 models and send each to one client. That is, we initialize clients’ local models by the sampled models. The accuracy is 71.2%, worse than FedBE (77.5%).
>
> Exp2: We investigate another idea. We performed FedBE for half of the 40 rounds. For the others, we perform FedAvg. We switch between them for every round, which reduces half of the distillation computation. We see 1% drop (76.5% vs. 77.5%). Both their accuracies are higher than no distillation in Exp1.
>
> These results justify the necessity of distillation. When server computation is a concern, we may switch between FedBE and FedAvg to trade accuracy for computation.
>
> **Table 1:** centralized training vs. FL: We have provided some discussions in our original submission. Please see section 5.2 Network depth at page 8 and section E.2. Unlike centralized training, in FL the client data are relatively small and may be non-i.i.d. Complex models are prone to overfitting and drift away from each other due to the non-i.i.d. data. For the compared networks in Table 1, the local models are trained to convergence (thus not under-trained), reaching 99+% local training accuracy at many rounds.
>
> **Section 5.1:** Thanks and we correct it: E = K*batch_size/|D_i|. We follow (Wang et al., 2020) to tune E based on the validation accuracy. We split 1K samples from the training data, and evaluate FedAvg after 40 rounds.
>
> **Footnote 2:** We provide two supports. In Figure 7, we perform FedAvg and show the test accuracy at every round, together with the accuracy by Bayesian ensemble. That is, we take the clients’ models (learned with FedAvg), construct the distribution, sample models, and perform ensemble. The Bayesian ensemble outperforms weight average at nearly all the rounds, even though it is noisy (i.e., not with 100% accuracy). Another evidence is in section D.2, in which we study one-round FL. The ensemble accuracy is 60+%, higher than 25% of FedAvg.
>
> **Appendix A:** We move section A to section 3 in the main text.
>
> **Approximated posterior:** For diagonal Gaussian, we follow SWAG (Maddox et al., 2019), which provides empirical results showing that the posterior spanned by SGD is approximately Gaussian. (Please see their section 4 and appendix A.) Diagonal Gaussians have also been used in other works for approximation (see the following).
>
> (Blundell et al.) Weight Uncertainty in Neural Networks. ICML 2015
>
> (Kirkpatrick et al.) Overcoming catastrophic forgetting in neural networks. PNAS 2017
>
> (Lee et al.) Deep Neural Networks as Gaussian Processes. ICLR 2018
>
> To justify our usage, we compare with SWAG. We perform FedBE on i.i.d CIFAR-10 with 10 clients for 40 rounds to obtain a global model. We then study two ways to obtain 10 models based on the global model and construct a diagonal Gaussian. (a) We train each client with 10 more iterations and apply Equation 5. (b) We aggregate clients’ data and run SWAG by collecting a model every 10 iterations. The KL divergence between the two Gaussians is 0.21. The two Gaussians are close.
>
> For Dirichlet, we do not explicitly approximate the posterior but are inspired by recent works that combine stochastic gradients for better models (see discussion above Equation 6). After all, we aim to construct a distribution to sample models for ensemble, and both methods are effective for this purpose.

---

### Author Response · Authors · 2020-11-22
**General responses to all the reviewers**

We thank the reviewers for their valuable feedback. We are encouraged that they found our method to be interesting (Reviewer 1 and 3), simple yet effective (Reviewer 4), novel (Reviewer 2, 3, 4: the Bayesian ensemble and SWA), and well-analyzed with extensive experiments (Reviewer 1, 3, 4). We are glad that Reviewer 4 found the type of work to be definitely needed, and Reviewer 1 found the problem we studied to be important. We are pleased that Reviewer 3 recognized our work as a drop-in replacement for weighted averaging. All the reviewers found our paper to be well-written.

*We address the common comment on computation below. We respond separately to the comments provided by each reviewer. We have incorporated some feedback into the revised version and will incorporate all feedback in the camera-ready version.*

**Computation:** We have some discussions and results (e.g., wall clock time of FedBE) in the original submission. Please be referred to section E.1 and the last paragraph of section 4. Using a 2080 Ti GPU on CIFAR-10 (ConvNet), building distributions and sampling models takes 0.2s, inference of one sampled model on the server data takes 2.4s, and distillation takes 10.4s, for each communication round. While constructing the ensemble predictions requires each sampled model to be evaluated on the unlabeled server data, as long as the server is computationally rich (e.g., with multiple GPUs), this step can be parallelized. In other words, while these computations can not be parallelized to the clients, it can be parallelized within the server.

In terms of computational complexity, let us denote by D the size of server data, S the number of involved clients per round, M the number of sampled models per round, and T the number of distillation epochs, then for each communication round, building distributions and sampling models takes O(S) and O(M),  inference of all models takes O(D*(S+M+1)), and distillation takes O(D*T).

We note that FedBE adds no extra computation to clients but only to the server. In many real-world applications, it is not unrealistic to assume that the server for federated learning has much richer computational resources than the clients. When computational time is a concern (e.g., no powerful servers), we have investigated a way in the response to Reviewer 3 to periodically perform FedBE (i.e., for some rounds, we perform FedBE; for others, FedAvg) to reduce the computational time with a slight drop on accuracy. To further save computation, we can reduce the epochs of distillation. While Figure 13 shows that reducing the epochs to below 10 would degrade the accuracy of FedBE, the accuracy is still much higher than FedAvg. We can also use fewer server data. Figures 3, 5, and 6 have shown that, even with just 1~2K server data, FedBE can already outperform FedAvg by a margin.

---

### Decision · Program_Chairs · 2021-01-07
**Final Decision**

**Decision:**

Accept (Poster)

**Comment:**

This is a well written paper with good experimentation.  The paper builds on the work of FedDF and does ablation studies to demonstrate its improvements.  The key original idea is the use of a common pool of unlabeled data which is used in transmitting partial results between local and global servers.  The results seem pretty good.

From a practical viewpoint, the unlabelled common data will, in most cases, need to be generated/artificial data since it will need to be public (to the other servers at least).  This option should be tested to demonstrate feasability.

AnonReviewer2 was concerned about whether it was fair to provide additional unlabelled data.  The authors tested this out and showed it was OK.  Regardless, the different servers could easily generate artificial data for this purpose.  AnonReviewer1 had a number of issues which the authors largely addressed. The other two reviewers appreciated the paper.  All reviewers gave constructive suggestions.